# Free Bilirubin Induces Neuro-Inflammation in an Induced Pluripotent Stem Cell-Derived Cortical Organoid Model of Crigler-Najjar Syndrome

**DOI:** 10.3390/cells12182277

**Published:** 2023-09-14

**Authors:** Abida Islam Pranty, Wasco Wruck, James Adjaye

**Affiliations:** 1Institute for Stem Cell Research and Regenerative Medicine, Medical Faculty, Heinrich-Heine University Düsseldorf, 40225 Düsseldorf, Germany; abida.pranty@hhu.de (A.I.P.); wasco.wruck@med.uni-duesseldorf.de (W.W.); 2Zayed Centre for Research into Rare Diseases in Children (ZCR), University College London (UCL)—EGA Institute for Women’s Health, 20 Guilford Street, London WC1N 1DZ, UK

**Keywords:** BIND, kernicterus, Crigler–Najjar syndrome, UGT1A1, 3D brain organoid, free bilirubin, neuro-inflammation

## Abstract

Bilirubin-induced neurological damage (BIND), which might progress to kernicterus, occurs as a consequence of defects in the bilirubin conjugation machinery, thus enabling albumin-unbound free bilirubin (BF) to cross the blood–brain barrier and accumulate within. A defect in the UGT1A1 enzyme-encoding gene, which is directly responsible for bilirubin conjugation, can cause Crigler–Najjar syndrome (CNS) and Gilbert’s syndrome. We used human-induced pluripotent stem cell (hiPSC)-derived 3D brain organoids to model BIND in vitro and unveil the molecular basis of the detrimental effects of BF in the developing human brain. Healthy and patient-derived iPSCs were differentiated into day-20 brain organoids, and then stimulated with 200 nM BF. Analyses at 24 and 72 h post-treatment point to BF-induced neuro-inflammation in both cell lines. Transcriptome, associated KEGG, and Gene Ontology analyses unveiled the activation of distinct inflammatory pathways, such as cytokine–cytokine receptor interaction, MAPK signaling, and NFκB activation. Furthermore, the mRNA expression and secretome analysis confirmed an upregulation of pro-inflammatory cytokines such as IL-6 and IL-8 upon BF stimulation. This novel study has provided insights into how a human iPSC-derived 3D brain organoid model can serve as a prospective platform for studying the etiology of BIND kernicterus.

## 1. Introduction

Neonatal jaundice is a common occurring transitional condition, caused by unconjugated hyperbilirubinemia, which affects about 85% of newborns in their first week of postnatal life [1,2]. Unconjugated hyperbilirubinemia is normally benign; nevertheless, the protective mechanisms of the brain can be surpassed in the infants when albumin-unbound free bilirubin (BF) starts accumulating and crosses the blood–brain barrier (BBB) due to defective bilirubin conjugation machinery. Elevated bilirubin levels result in severe brain injury including short-term and long-term neurodevelopmental disabilities, which can progress into acute or chronic bilirubin encephalopathy, known as kernicterus, or bilirubin-induced neurological dysfunction (BIND) [3]. Kernicterus is a complex neuropathological ailment, which can lead to acute or chronic neurological disabilities, resulting in BIND [4,5]. A genetic disorder caused by a defective UGT1A1 (UDP glucuronosyltransferase family 1 member A1) enzyme leads to Crigler–Najjar syndrome (CNS) and can also cause BIND [6]. Bilirubin-UGT1 (UGT1A1) is the only isoform of the uridine 5′-diphospho-glucuronosyltransferases (UGTs) enzyme family that specifically contributes to bilirubin conjugation. Therefore, a mutation in the gene encoding UGT1A1 can cause CNS type 1 and 2 by complete or partial inactivation of the enzyme, respectively, which prevents the liver from metabolizing bilirubin [7]. The more severe type (CNS-1) is characterized by total deficiency of the UGT1A1 enzyme, while CNS-2 (the milder one) is characterized by a partial UGT1A1 deficiency [6]. A decrease in or loss of UGT1A1 activity due to hereditary defects in UGT1A1 hinders bilirubin conjugation, causing BF to accumulate in serum, cross the BBB, and eventually get deposited in the basal ganglia or cerebellum, thereby resulting in BIND [7,8,9]. The mechanism underlying BIND and the correlation between the BF levels and neurological abnormalities are not well understood.

Neuro-inflammation is one of the key features observed in the distinct model systems of bilirubin-induced brain toxicity. It has been shown that in *UGT1A1* knocked-out mice, Toll-like receptor 2 (TLR2) is required for regulating gliosis, pro-inflammatory mediators, and oxidative stress when neonatal mice are exposed to severe hyperbilirubinemia, but TLR2 also has anti-apoptotic properties. This correlates with the upregulation of tumor necrosis factor alpha (TNF-α), interleukin (IL) 1ß, and IL-6, thus indicative of neuro-inflammation in the CNS [10]. Neuronal damage in distinct neurodegenerative diseases and neuro-inflammatory diseases (such as Alzheimer’s disease, Parkinson’s disease, and Multiple Sclerosis) are reported to be mediated by various inflammatory and neurotoxic factors, such as IL-6, IL-8, TNF-α, intracellular Ca^2+^ elevation, chemokines, activation of mitogen-activated protein kinases (MAPKs), and nuclear factor kappa B (NFκB) [11]. TNF-α and NFκB are the key mediators of bilirubin-induced inflammatory responses in murine models [12]. Sustained exposure of developing mouse cerebellum with free unconjugated bilirubin induces activation of oxidative stress, endoplasmic reticulum (ER) stress, and inflammatory markers, thus pointing to inflammation as a key contributor of bilirubin-induced damage in conjunction with ER stress in the onset of neurotoxicity [12]. Immature rat neurons also manifested distinct features of oxidative stress and cell dysfunction upon bilirubin exposure by increased ROS production, the disruption of the glutathione redox status, and cell death [13]. Bilirubin-induced apoptosis and necrosis-like cell death leading to neuritic atrophy and astrocyte activation have also been observed in monotypic nerve cell cultures [14,15]. During necrosis, various receptors become activated and the uncontrolled release of cytosolic constituents may induce inflammatory responses in the surrounding tissue [16]. Various stimuli (e.g., cytokines) can cause both apoptosis and necrosis in the same cell population. Furthermore, by modulating signaling pathways (such as death receptors and kinase cascades), it is possible to switch between apoptosis and necrosis [16,17]. Necroptosis is a programmed form of necrosis, or inflammatory cell death, which allows the cell to undergo cellular suicide in a caspase-independent manner in the presence of caspase inhibitors [18]. Elevated bilirubin levels perturb the plasma, mitochondrial, and/or ER membranes of neuronal cells, probably leading to neuronal excitotoxicity, mitochondrial energy failure, or increased intracellular calcium concentration [Ca^2+^], which are assumed to be linked to the pathogenesis of BIND. Increased [Ca^2+^] and the following downstream events may activate proteolytic enzymes, apoptotic pathways, and/or necrosis, depending on the intensity and duration of the bilirubin exposure [15].

In this study, we used human-induced pluripotent stem cell (hiPSC)-derived 3D brain organoids as a potential in vitro model system to enhance our understanding of BIND-associated molecular pathogenesis and the breadth of complexities at the cellular and molecular levels that accompany the detrimental effects of free bilirubin to the developing human brain. hiPSC derived from a healthy and a CNS patient were differentiated into day-20 brain organoids and then continuously stimulated with 200 nM BF to observe the possible short-term and long-term BF-induced effects [3,19]. We performed further analyses and observed the induction of neuro-inflammation along with the activation of the cytokine–cytokine receptor interaction, calcium signaling pathway, MAPK signaling pathway, and neuroactive ligand–receptor interaction as processes leading to BIND.

## 2. Materials and Methods

### 2.1. Cell Cultivation, Formation of Neural Cortical Organoids, and Bilirubin Treatment on Cortical Organoids

The hiPSC lines derived from the renal progenitor cells isolated from the urine of a 51-year-old healthy male of African origin (UM51) and fibroblast cells from a male Crigler–Najjar syndrome (CNS) patient (i705-C2) were used in this study [20,21] (Appendix A). The cells were plated on Matrigel (Corning, New York, NY, USA)-coated culture dishes using mTeSR plus medium (StemCell Technologies, Vancouver, Canada). The cultures were routinely tested for mycoplasma contamination. The cells were dissociated into small aggregates with ReLeSR (StemCell Technologies, Vancouver, Canada) every 5–7 days and split in a 1:5 ratio into fresh Matrigel-coated dishes. Alternatively, the cells were also split as single cells using accutase (Life Technologies, Waltham, MA, USA) while seeding for organoid generation. The protocol described by Gabriel et al., 2016, was employed to differentiate the iPSCs into cortical organoids with minor modifications [22]. Briefly, 20,000 single iPSCs were seeded onto each well of a U-bottom 96-well plate (NucleonTM SpheraTM, Thermo Fisher Scientific, Rockford, IL, USA) to form embryoid bodies (EBs) with mTesR plus medium and 10 µM ROCK inhibitor Y-27632 (Tocris Bioscience, Wiesbaden, Germany). The EBs were cultured on the plate for 5 days with the neural induction medium (StemCell Technologies, Vancouver, Canada) to initiate neural induction. On day 6, the EBs were transferred into a bioreactor (PFIEFFER, Lahnau, Germany) with a differentiation medium consisting of DMEM/F12 and the Neural Basal Medium (in 1:1 ratio), supplemented with 1:200 N2, 1:100 L-glutamine, 1:100 B27 w/o vitamin A, 100 U/mL penicillin, 100 mg/mL streptomycin, 0.05 mM MEM non-essential amino acids (NEAA), 0.05 mM β-mercaptoethanol (all from Gibco, Waltham, MA, USA), and 23 µM insulin (Sigma, Taufkirchen, Germany) (see Appendix A). The spinner flasks were coated with anti-adherent rinsing solution (StemCell Technologies, Vancouver, Canada) before transferring the EBs. The transferred EBs were counted as day-0 organoids from this time point (differentiation day 6), as the spheres were transferred into the spinner flask to spontaneously pattern into cortical organoids. From day 9 onward, 0.5 µM of dorsomorphin (Tocris Bioscience, Wiesbaden, Germany) and 5 µM SB431542 (Tocris Bioscience, Wiesbaden, Germany) were added to the differentiation medium, and the medium in the bioreactor was changed once a week. The day-20 cortical organoids were treated up to 72 h with 200 nM free bilirubin (BF), while dimethyl sulfoxide (DMSO) served as the control and the medium was refreshed every 24 h. In brief, free bilirubin (Sigma-Aldrich Chemicals, Taufkirchen, Germany) was used in this study. DMSO was used to dissolve the bilirubin to obtain a stock concentration of 100 mM and further diluted to 200 µM. For the treatment and control conditions, BF and DMSO were diluted (1:1000) in the culture medium, respectively.

Hepatocyte-like cells (HLCs) were derived from the iPSCs following the protocol described by Graffmann et al., 2016 (Appendix A) [23].

### 2.2. Cryosectioning

The cells (HLCs) were fixed in 4% paraformaldehyde (PFA) (Polysciences, Warrington, FL, USA) for 10 min and the cortical organoids were fixed for 30 min at 37 °C. After washing with PBS, the cells were directly used for staining and the organoids were dehydrated with 30% sucrose in PBS overnight at 4 °C. Then, the organoids were embedded using the Tissue-Tek OCT Compound (embedding medium) (Sakura Finetek, Umkirch, Germany) in cryo-molds and snap-frozen in 2-methylbutan (Carl Roth, Karlsruhe, Germany) and dry ice. The embedded organoids were stored at −80 °C. The organoids were sectioned into 15 µm sections using a Cryostat (CM1850, Leica, Nussloch, Germany) and captured in Superfrost plus slides (Thermo Scientific, Waltham, MA, USA). The sectioned organoid slices were stored at −80 °C prior to the immunofluorescence-based analyses.

### 2.3. Immunocytochemistry

The fixed cells were permeabilized for 10 min with 0.1% Triton X-100 in PBS+Glycine (30 mM Glycine) at room temperature (RT). They were then washed once with PBS and then the unspecific binding sites were blocked for 2 h at RT with blocking buffer 0.3% BSA in PBS+Glycine. The frozen sections were thawed at RT and PBS was added drop by drop without touching the organoid sections and incubated at RT for 15 min. The Tissue Tek was washed off with PBS and the slide was washed once more with PBS. The sections were permeabilized with 0.7% Triton X-100 + 0.2% Tween 20 in PBS+Glycine for 15 min at RT. After, permeabilization blocking was carried out for 2 h at RT with 0.2% Triton X-100 + 0.3% BSA in PBS+Glycine. For the cells and organoid sections, the primary antibody solution was incubated overnight at 4 °C (see Appendix A). After removing the primary antibodies and thorough washing, the secondary antibodies were added for 2 h and incubated at RT. The nuclei were stained with Hoechst. The stained cells and sections were imaged using a Zeiss fluorescence microscope (LSM 700). Particular staining regions were observed under a Zeiss confocal microscope (LSM 700). Individual channel images were processed and merged with ImageJ software version 1.53c.

### 2.4. TUNEL Assay

Apoptotic cells were detected using the DeadEnd™ Fluorometric TUNEL System (Promega, G3250, Madison, WI, USA) following the manufacturer’ s protocol.

### 2.5. Reverse Transcriptase PCR (RT-PCR)

The cells (HLCs) and cortical organoids were lysed in Trizol to isolate the RNA. In total, 7–8 BF-treated and non-treated organoids were taken for RNA isolation. The RNA was isolated with the Direct-zol™ RNA Isolation Kit (Zymo Research, Freiburg, Germany) according to the user’s manual, including the 15 min and 30 min DNase digestion step for cells and organoids, respectively. A total of 500 ng of RNA was reverse-transcribed using the TaqMan Reverse Transcription Kit (Applied Biosystems, Waltham, MA, USA). The primer sequences are shown in Appendix A. Real-time PCRs were performed in technical and independent experiment triplicates (n = 3) with Power Sybr Green Master Mix (Life Technologies, Darmstadt, Germany) on a VIIA7 (Life Technologies, Darmstadt, Germany) machine. The mean values were normalized to the ribosomal protein lateral stalk subunit P0 (*RPLP0*) and the fold change was calculated using the indicated controls. The observed fold changes are depicted as mean values with a 95% confidence interval (CI). A statistical analysis of the data was conducted by using Student′s unpaired two-sample *t*-test on the basis of the difference between each sample mean compared to the corresponding control mean.

### 2.6. Human XL Cytokine Assay

The conditioned medium or supernatant of the control and BF-treated cortical organoids from both the 24 and 72 h treatments were stored and used for the proteome profiler antibody array. The relative expression levels of 105 soluble human proteins and cytokines were determined using the Human XL Cytokine Array Kit from R&D Systems. The cytokine array was performed following the manufacturer’s guidelines. In brief, the membranes were blocked for 1 h on a rocking platform using the provided blocking buffer and then the samples were prepared by diluting the desired quantity to a final volume of 1.5 mL with the distinct array buffer (array buffer 6). The sample mixtures were pipetted onto the blocked membranes and were incubated overnight at 4 °C on a rocking platform. The membranes were then washed three times with washing buffer for 10 min each at RT. Then, the membranes were incubated with the detection antibody cocktail for 1 h at RT and then washed three times thoroughly. Afterward, Streptavidin-HRP was added onto the membranes, which were incubated for 30 min at RT. The ECL detection reagent (Cytiva, Freiburg, Germany) was used to visualize the spots on the membrane and then detected in a Fusion FX instrument (PeqLab, Erlangen, Germany).

### 2.7. Image and Data Analysis of the Human XL Cytokine Array

After performing the cytokine array with non-treated and BF-treated organoids, the hybridizations of the cytokine arrays were scanned with the Fusion FX instrument (PeqLab, Erlangen, Germany) and read into the FIJI/ImageJ software version 1.53c, Java 1.8.0_172, where the spots were quantified as described in our previous publication by Wruck et al. [24,25]. The spots were associated with the cytokine identifiers provided by the manufacturer (Proteome Profiler Array from R&D Systems, Human XL Cytokine Array Kit, Catalog Number ARY022B). The integrated densities of the spots as a result of the quantification were read into the R/Bioconductor [26]. The Robust Spline Normalization from the R/Bioconductor package lumi was applied to the data [27]. The expressed and differentially expressed cytokines were determined as described previously [24]. The differential expression was tested via the moderated *t*-test from the Bioconductor package limma, adjusted for the false discovery rate via the method of Benjamini and Hochberg [28,29]. The cytokines expressed in both conditions (detection *p* value < 0.05) with an adjusted limma differential expression *p* value < 0.05 were considered upregulated when their ratio was greater than 1.2 (6/5) and downregulated when their ratio was less than 0.8333 (5/6).

### 2.8. Analysis of Gene Expression Data

The i705-C2 and UM51 organoid samples, untreated and treated with BF, were measured after 24 and 72 h on the Affymetrix Human Clariom S Array at the core facility of the Heinrich-Heine Universität, Düsseldorf (BMFZ: Biomedizinisches Forschungszentrum). Processing of the microarray data was performed in the R/Bioconductor environment [26]. The background correction and normalization with the Robust Multi-array Average (RMA) method was achieved by the Bioconductor package oligo [30]. Using the values of the dedicated background spots on the microarray, a statistic was calculated to determine a detection p value to judge if a probeset was expressed (detection *p* < 0.05) as was described before in Graffmann et al. [23]. The probesets expressed following this criterion were mapped to unique gene symbols according to the annotations provided by Affymetrix and compared in Venn diagrams via the VennDiagram package [31]. The function heatmap.2 from the R gplots package was applied to draw the heatmaps and the associated clustering dendrograms using the Pearson correlation as the similarity measure and color-scaling by Z-scores of rows (genes) [32]. The genes with a detection *p* value below 0.05 in both conditions were considered upregulated when the ratio was greater than 1.5 or downregulated when the ratio was less than 0.67.

### 2.9. Analysis of Pathways and Gene Ontologies (GOs)

For the analysis of the KEGG (Kyoto Encyclopedia of Genes and Genomes) pathways, the associations between the genes and pathways were downloaded from the KEGG website on 6 July 2020 [33]. The over-representation of pathways in the gene sets of interest was tested via the R-built in the hypergeometric test. The R package GOstats was applied for calculating over-represented gene ontologies [34]. For the dot plots of the most significantly over-represented pathways, the R package ggplot2 was employed [35].

### 2.10. Western Blotting

The total proteins from the cortical organoids and HLCs were isolated using the RIPA buffer (Sigma-Aldrich Chemicals, Taufkirchen, Germany), consisting of protease and phosphatase inhibitors (Roche, Mannheim, Germany). Afterward, the Pierce BCA Protein Assay Kit (Thermo Fisher, Waltham, MA, USA) was used to determine the protein concentrations. Approximately 20 µg of the heat-denatured protein lysate of each sample was loaded on a 4–12% SDS-PAGE and then transferred by wet blotting onto a 0.45 µm nitrocellulose membrane (GE healthcare, Solingen, Germany). After 1 h of blocking with 5% milk in TBST, the membranes were stained with anti-P53, anti-γH2AX, anti-UGT1A1, anti-CREB, anti-phospho-CREB, and anti-phospho-P38MAPK antibodies. Incubation with primary antibodies was performed overnight at 4 °C. After washing the membranes three times with TBST, the secondary antibody incubation was performed for 2 h at RT followed by washing with TBST (Appendix A). Anti β-Actin and anti-GAPDH (glyceraldehyde-3-phosphate dehydrogenase) were used as the housekeeping proteins to normalize the protein expression. ECL Western Blotting Detection Reagents (Cytiva, Freiburg, Germany) were used to visualize the stained protein bands and then detected in a Fusion FX instrument (PeqLab, Erlangen, Germany). A band intensity quantification and analysis was performed with Fusion Capt Advance software FX7 16.08 (PeqLab, Erlangen, Germany) and was normalized to the β-Actin band intensity.

### 2.11. Statistical Analysis

Statistical analyses for the comparison of each sample to their corresponding control were carried using Student’s unpaired two-sample *t*-test. The calculations were performed with GraphPad Prism Software version 8.0.2 (263) (GraphPad software, San Diego, CA, USA) and Microsoft Excel. An asterisk depicts significance, which is determined by a *p* value ≤ 0.05, ** *p* < 0.01 and *** *p* < 0.001. The error bars depict the ± 95% confidence interval (qRT-PCR data) or mean ± standard deviation (SD) (IF quantification and WB).

### 2.12. Measurement of Cytochrome P450 Activity

The P450-GloTM CYP2D6 Assay and P450-GloTM CYP3A4 Assay Luciferin-IPA (Promega) kits were used to measure the Cytochrome P450 2D6 and P450 3A4 activity by employing a luminometer (Lumat LB 9507, Berthold Technologies, Bad Wildbad, Germany).

## 3. Results

### 3.1. iPSC-Derived Cortical Organoids Show Typical Cortical Neuronal Features

The hiPSCs derived from the renal progenitor cells isolated from the urine of a 51-year-old healthy male of African origin (UM51) and the fibroblast cells from a Crigler–Najjar syndrome (CNS) male patient (i705-C2) were cultured as colonies and used to generate 3D cortical neuronal organoids in triplicate. The i705-C2 iPSCs-derived HLCs retained defective UGT1A1 expression, as the cells were derived from the Crigler–Najjar syndrome (CNS) patient (Appendix A) [20,21]. The iPSCs were seeded as single cells to form embryoid bodies (EBs) and then transferred into a bioreactor to grow spontaneously as organoids (Figure 1a,b). Similar to previously established cerebral organoids, our generated organoids recapitulated human cortical developmental features with progenitor zone organization and the presence of radial glia stem cells and cortical neurons (Figure 1c) [36]. The self-patterned organoids showed cortical neuronal identity with the expression of the radial glia marker-SRY-box transcription factor 2 (SOX2), neuronal markers Beta III tubulin (TUJ1), microtubule-associated protein 2 (MAP2), and doublecortin (DCX) at day 15 (Figure 1c). These generated cortical organoids were treated with BF for further experimental analysis (Figure 1a).

### 3.2. Bilirubin (BF)-Induced Neuro-Inflammation with Elevated Expression of Pro-Inflammatory Cytokines

The day-20 cortical organoids were treated continuously with 200 nM free bilirubin for 72 h in the bioreactor (Figure 1a). The RNA from the UM51 and i705-C2 cortical organoids was isolated in triplicate (n = 3) at the 24 h and 72 h time points of the BF-treated and control conditions to investigate the expression of pro-inflammatory-associated cytokines at the mRNA level. In parallel, the supernatant was collected at both time points to carry out a further secretome analysis using cytokine arrays (Appendix A). The qRT-PCR analysis from the i705-C2 line showed a 3-fold increase in the *IL-6* and *IL-8* expression at 24 h post-treatment but decreased to 2-fold at 72 h, while the *IL-8* gene expression was enhanced 3-fold also for the UM51 healthy control line (Figure 2a). The *IL-6* mRNA expression was enhanced 1.5-fold only at 72 h post-treatment in the healthy line. On the other hand, the expression of *TNF-α* did not show any change upon BF exposure for the UM51 line, but a slight increase with a 1.5-fold (24 h and 72 h) change was observed in the i705-C2 line. Additionally, a heatmap obtained from the cytokine array-based analysis showed the secretome profile of various cytokines after BF treatment for both cell lines at 24 h and 72 h (Figure 2b) (Appendix A). The enhanced secretion of the pro-inflammatory cytokines such as IL-6, IL-8, TNF-α, IL1-ß, IL-16, and INF-G was observed along with the anti-inflammatory cytokine interleukin (IL)-1 receptor antagonist (IL1-ra), IL-4, IL-11, and IL-13 secretion (Figure 2b). A number of the major anti-inflammatory cytokines which regulate pro-inflammatory cytokine expression include IL1-ra, IL-4, IL-10, IL-11, and IL-13 [37]. We could see a slight decrease in the IL1-ra, IL-10, IL-11, and IL-13 secretion in the 24 hrs post-treatment condition, while IL-4 expression was slightly increased. On the other hand, at 72 h, the expression of IL-4 decreased whilst that of IL1-ra, IL-10, and IL-13 showed modest increases in the treated condition (Appendix A). The secretome analysis also revealed enhanced vascular endothelial growth factor (VEGF) secretion and reduced sex hormone-binding globulin (SHBG) secretion upon BF treatment for all the conditions, except UM51 at 24 h (Figure 2b) (Appendix A). The healthy line showed a 2-fold increase in the *VEGF* mRNA expression at 72 h, whilst the patient line showed a 1.4-fold and 1.6-fold increase at 24 and 72 h, respectively (Appendix A). On the other hand, the *SHBG* expression was slightly decreased by 0.6-fold (UM51) and 0.8-fold (i705-C2) at 72 h post-treatment (Appendix A). Overall, the increase in both the mRNA expression and protein secretion of IL-6, IL-8, and TNF-α implies the initiation of neuro-inflammation in the cortical organoids upon BF treatment.

### 3.3. Distinct Inflammation-Associated Pathways Are Activated by BF in Cortical Organoids

To further investigate the neuro-inflammatory effect of BF on cortical organoids at the molecular level, the RNA was isolated from 24 and 72 h post-treated and untreated (control) UM51 and i705-C2 cortical organoids for a microarray-based transcriptome analysis. The Venn diagrams obtained from the transcriptome analysis revealed that the i705-C2 cortical organoids expressed 14,721 genes in common between the treatment and control and uniquely expressed 906 genes at 24 h post-treatment, whereas at 72 h, 15,317 genes were expressed in common, and 124 genes were uniquely expressed in the i705-C2 organoids (Figure 3a). On the other hand, the UM51 cortical organoids expressed 15,491 and 15,731 common sets of genes, while 289 and 351 genes were uniquely expressed upon BF exposure at 24 h and 72 h, respectively (Figure 3a). The KEGG pathways associated with the uniquely expressed genes at 24 h post-treatment included *TNFSF12* (TNF superfamily member 12), *AZI2* (5-azacytidine induced 2)*,* and *MyD88* (myeloid differentiation primary response protein), and, for example, a cytokine–cytokine receptor interaction was activated in the i705-C2 line, which gradually decreased at 72 h (Figure 3b,c) (Appendix A) (Appendix A). Cytokine–cytokine receptor activation was not observed in the healthy UM51 line at 24 h but at 72 h (Figure 3d) (Appendix A). However, the calcium and MAPK signaling pathways and the neuroactive ligand–receptor interactions were activated at 24 h for the patient line and for both cell lines at 72 h post-BF exposure (Figure 3b–d) (Appendix A).

### 3.4. BF Treatment Differentially Regulates DNA Damage and Repairs-Related Pathways

The transcriptome analysis of the cortical organoids provided an overview of the GO terms and associated KEGG pathways of the differentially regulated genes (common set of genes) upon BF exposure (Appendix A). The KEGG pathways revealed that DNA damage and repair-related pathways such as P53 signaling, homologous recombination, and the Fanconi Anemia pathway were upregulated in the UM51 line and downregulated in the i705-C2 line at 24 h (Figure 4(bii,biii)) (Appendix A). In parallel, at 24 h, the i705-C2 line showed activation of the NFκB, PI3K, and chemokine signaling pathways, indicating to the initiation of inflammation, while the cellular developmental processes related to the Notch and TGFβ signaling pathways were upregulated (Figure 4(bi)). Interestingly, a number of these inflammatory and development-related pathways (neuroactive ligand–receptor interaction, cAMP signaling pathway, and cytokine–cytokine receptor interaction) were observed to be downregulated at 24 h in the UM51 line (Figure 4(biv)). The mRNA expression of *NLRP3* (NLR family pyrin domain-containing 3) seems to be enhanced only for the i705-C2 line, which is an upstream activator of NFκB signaling and plays a role in inflammation, immune response, and apoptosis. The activation and increased *NLRP3* mRNA expression might indicate NFκB-mediated inflammatory responses upon BF exposure (Appendix A). UM51 showed a slight increase in both the mRNA (1.2-fold) and protein expression of the cAMP-response element-binding protein (CREB) (1.3-fold) and the protein expression for phospho-CREB (1.13-fold), while the i705-C2 line showed only slightly enhanced mRNA expression (Appendix A). Moreover, the UM51 line showed slightly enhanced mRNA expression of the DNA damage–repair and apoptosis-related genes at 24 and 72 h such as *P53* (tumor protein P53) (1.24-fold and 1.14-fold), *BCL2* (B-cell lymphoma 2) (1.5-fold and 1.3-fold), *ATM* (ataxia telangiectasia mutated) (1.41-fold), *ATR* (ataxia telangiectasia and Rad3-related protein) (1.3-fold and 1.3-fold), *CHEK1* (checkpoint kinase 1) (1.3-fold and 1.2-fold), and *CHEK2* (1.6-fold) compared to the i705-C2 line (Figure 4(ai)). However, the patient line showed increased *BCL2* (1.3-fold) and *MDM2* (mouse double minute 2 homolog) (1.4-fold) expression at 24 h and increased *P53* (1.6-fold) expression at 72 h post-BF treatment. In addition to that, the Western blot (WB) analysis revealed a 1.64-fold increase in the P53 protein expression at 24 h in the healthy line and with a non-significant 1.1-fold increase at 72 h in the patient line (Figure 4(aii,iii)). Furthermore, the GO terms of the common set of expressed genes revealed neurodevelopmental pathways to be differentially upregulated with the activation of the cellular developmental process, nervous system development, axon development, axon guidance, tight junction, positive regulation of axogenesis, and positive regulation of synapse assembly for both cell lines at 72 h post-BF treatment (Appendix A) (Appendix A).

### 3.5. BF Induces Apoptotic Cell Death in Cortical Organoid

The apoptotic cell death in the BF-treated cortical organoids was evaluated by immunofluorescence and Western blot analysis. Both cell lines exhibited an increased level of cell death based on terminal deoxynucleotidyl transferase dUTP nick end labeling (TUNEL) at 24 h post-treatment (Figure 5a,b). Cleaved caspase 3 (CASP 3) positive cells were observed in the organoid sections, pointing at apoptotic cell death at 24 h post-treatment (Figure 5a,b).

Immunofluorescence-based quantification revealed a 34% increase in the TUNEL and cleaved Caspase 3 positive cells in the UM51 organoid sections, whereas the i705-C2 sections had 22% TUNEL positive and 25% cleaved Caspase 3 positive cells at 24 h (Figure 5c). The γH2AX expression, which indicates both DNA damage or apoptotic cell death, showed an enhanced protein expression of 1.26-fold based on the WB analysis on the healthy line after 24 h of BF exposure, whilst the patient line showed almost no change in the γH2AX expression (1.04-fold increase) (Figure 5d). The immunofluorescence-based staining also showed a 15% increase in the γH2AX expression at 24 h post-BF treatment for the healthy line (Appendix A). A further indication of DNA damage was observed with the upregulation of the P53-mediated signaling pathway in the KEGG analysis of the UM51 transcriptome at 24 h (Figure 4(biii)). The necroptosis-associated genes shown in the Pearson heatmap derived from our transcriptome analyses revealed that the upregulated expressed genes for the patient organoid (X-linked inhibitor of apoptosis protein (*XIAP*); tumor necrosis factor receptor-associated factors (*TRAF*), *TRAF2* and *TRAF5*; signal transducers and activators of transcription (*STAT*), *STAT1*, *STAT2*, *STAT3*, and *STAT4*; janus kinase 2 (*JAK2*); and baculoviral IAP repeat containing (*BIRC*), *BIRC3*, *BIRC5*, and *BIRC6*) are associated with inflammation, whilst the upregulated genes (*FAS* and *CASP8*) for the healthy organoid are associated with apoptosis (Appendix A). Regardless of the BF treatment, the transcriptomes of the patient samples were clustered separately in the heatmap from the healthy line, which might also indicate that the observed upregulation is a phenotype of the defective UGT1A1 in the patient line-derived organoids.

## 4. Discussion

In this study, we generated iPSC-derived 3D cortical organoids to model BIND in vitro and unveil insights into the detrimental effects of BF in the developing human brain at the molecular and protein level. Our in vitro model comprises a healthy iPSC (UM51) and a CNS patient-derived (i705-C2) iPSC, harboring the UGT1A1 mutation [20,21]. Defective UGT1A1 protein expression was confirmed after differentiation of the iPSCs into hepatocyte-like cells (HLCs).

BF can interfere BBB integrity by glutathione disruption and increased endothelial nitric oxide synthase (NOS) by enhanced cytokine release [3,38]. Cytokines play a key role in regulating nerve cell responses during a brain injury [39]. Cytokines might have both beneficial and detrimental effects on neurons depending on their levels of secretion [4,40,41]. The production of pro-inflammatory mediators can cause neuronal apoptosis and neuro-inflammation [42,43]. Even though microglia are the key cell type in the central nervous system which secrete pro-inflammatory cytokines upon stress, bilirubin-treated neurons showed enhanced secretion of IL-6 with decreased secretion of IL-1β [44]. Based on these findings, we analyzed the mRNA expression of several pro-inflammatory cytokines and their secretome profile. The mRNA expression from the control and BF-treated conditions showed a 3-fold increase in the *IL-6* and *IL-8* expression at 24 h post-treatment in the i705-C2 line which then decreased to 2-fold at 72 h. A similar pattern was observed in the secretome analysis for the i705-C2 line with an increased secretion of IL-6 and IL-8 at 24 h post-treatment, which then decreased at 72 h. The secretion of IL-6 can be repressed by IL-10 [37]. The increased secretion levels of the anti-inflammatory proteins IL-10 and IL-13 at 72 h might repress the IL-6 and IL-8 secretions. This observation might imply that inflammatory responses were initiated earlier in the i705-C2 organoids and gradually reverted to normal levels with time as a consequence of the cellular defense mechanisms establishing homeostasis. On the other hand, the UM51 (healthy) line showed 3-fold enhanced *IL-8* mRNA expression only at 24 h, while the IL-8 protein secretion was decreased at this time point. However, the IL-8 protein secretion was increased at 72 h. In parallel, the IL-6 protein secretion was enhanced for both 24 and 72 h post-BF treatment, while the mRNA expression showed a 1.5-fold increase only at 72 h. The *TNF-α* levels did not show any elevation upon BF exposure for the UM51 line but a 1.5-fold change increase in both the secretome and mRNA expression in the i705-C2 line. Fernandes et al. previously reported that bilirubin enhances the tumor necrosis factor receptor 1 (TNFR1) protein level in neural cells (such as astrocytes) along with a time-dependent release of TNF-α, IL-1ß, and IL-6. But neurons secrete merely low levels of IL-6 and even to a lesser extent TNF-α [4]. The correlation of the mRNA and secretome expression observed in our results suggests that mRNA and protein expression might have a slight time shift in their expression as a response to stress to induce inflammation in CNS. For example, the patient organoids showed increased IL1-β secretion only at 24 h, while the IL1-ra secretion was increased at 72 h for both cell lines. IL1-ra is a natural inhibitor of IL1-β, as this receptor competitively binds to the same receptor as IL1-β. Consequently, IL1-β-mediated cellular changes are obstructed, which might be the reason for the observed decrease in IL1-β secretion at 72 h post-BF treatment [37,45,46]. Having the anti-inflammatory effects, increased IL1-ra might antagonize cytokine-mediated inflammatory responses to re-establish the normal condition at 72 h. These observations point to a variation in the levels and patterns of initiation and responses to inflammation post-BF treatment. The secretome analysis also revealed enhanced VEGF and reduced SHBG secretion upon BF treatment for all conditions, except UM51 at 24 h. Previous studies described low VEGF expression in the adult human brain and upregulated VEGF levels have been observed in chronic neuro-inflammation [47,48]. Increased VEGF expression might be a cause or response to bilirubin treatment, which remains a question because of the multiple roles played by VEGF. On the other hand, SHBG exhibits anti-inflammatory effects in macrophages and adipocytes; however, SHBG expression in the brain is not yet known [49]. A slight reduction in *SHBG* expression by 0.6-fold (UM51) and 0.8-fold (i705-C2) at 72hrs post-BF treatment might indicate a tendency toward the initiation of an anti-inflammatory response. The secretome profile unveiled a pro-inflammatory CNS environment with variable cytokine secretion for both the healthy and disease cell lines at different time points, thus indicating pronounced neuro-inflammation upon BF treatment. Overall, in accord with Brites et al., the increase in mRNA expression and secreted levels of IL-6, IL-8, and TNF-α along with other inflammatory cytokines points to the onset or initiation of neuro-inflammation in the cortical organoids upon BF treatment [3,4,50].

Next, we performed a transcriptome-based microarray analysis to have an overview of the molecular effects of BF on cortical organoids, such as differential gene expression and associated biological processes (GO-BP) and KEGG pathways. Both cell lines showed activation of the cytokine–cytokine receptor interaction, calcium-signaling pathway, MAPK signaling, and neuroactive ligand–receptor interaction, among their uniquely expressed gene sets. In the patient line, cytokine–cytokine receptor activation was activated at both 24 and 72 h in the BF-treated condition, whereas in the healthy line this pathway seemed to be activated at 72 h. The observed GO terms and KEGG-associated pathways did not show activation of any of these mentioned pathways at 24 h in the treated condition in the UM51 line. Activation of GO-BP inflammatory bowel disease was observed in the i705-C2-24 h and UM51-72 h post-treated conditions. These findings point toward a possibility that the patient line shows an accelerated response to initiate inflammatory responses (24 h in this case) than the healthy line. The slightly enhanced mRNA expression of *TNSF12* (1.12-fold in UM51 and 1.6-fold in i705-C2), *AZI2* (1.2-fold in i705-C2), and *MyD88* (1.2-fold in i705-C2) was observed upon BF treatment. The increased expression of these genes points to the initiation of inflammation upon BF treatment [51,52,53,54,55].

The GO-BP terms revealed at 72 h the activation of the positive regulation of NFκB transcription factor activity (*AR*, *TRADD*, and *RPS6KA4*) in the i705-C2 line, while the positive regulation of the CREB transcription factor activity (*RELN* and *RPS6KA4*) and NLRP3 inflammasome complex assembly (*GBP5* and *NLRP3*) was revealed in the UM51 line. The activation and increased *NLRP3* mRNA expression might indicate NFκB-mediated inflammatory responses upon BF exposure. Similar to astrocytes and microglia, bilirubin-induced NFκB activation was observed in neurons as well but at lower levels [11,28]. Based on the cellular context, NFκB plays diverse roles in the central nervous system. The axonal growth of neurons can be implicated with NFκB activation [56,57]. NFκB may also regulate neural development, plasticity, and neurogenesis [4,58,59]. From our observed results, the GO-BP terms of the common set of genes revealed the neurodevelopmental pathways to be upregulated for both cell lines at 72 h post-BF treatment with the activation of nervous system development, axon development, axon guidance, tight junction, positive regulation of axogenesis, and synapse assembly. On the other hand, activation of the MAPK cascade was also observed in both the UM51 and i705-C2 lines upon BF treatment. MAPK signaling pathways were observed to be activated in bilirubin-treated astrocytes as well [4]. A slight enhancement with a 1.25-fold increase in the phospho-P38 protein expression was observed for the patient line post-BF exposure by Western blot analysis.

The KEGG pathways associated with the common set of genes revealed that DNA damage and repair-related pathways such as P53 signaling, homologous recombination, and Fanconi anemia pathway were upregulated in the UM51 line at 24 h. These pathways were downregulated at 24 h in the i705-C2 line. With respect to DNA damage, P53 induces multiple classes of its target genes, such as metabolic genes, DNA repair genes, cell-cycle arrest, and cell death effectors [60]. The UM51 line showed enhanced mRNA expression of the DNA damage–repair and apoptosis-related genes such as *P53*, *BCl2*, *ATM*, *ATR*, *CHEK1*, and *CHEK2* compared to the i705-C2 line. The P53 protein expression was 1.64-fold upregulated at 24 h in the healthy line and the level decreased at 72 h. Conversely, there was no increase in the P53 protein expression (1.1-fold at 72 h) in the patient line, but an increase in the *P53* mRNA expression (1.6-fold) was observed at 72 h. Interestingly, a number of inflammatory and development-related pathways (neuroactive ligand–receptor interaction and cytokine–cytokine receptor interaction) were downregulated at 24 h in the UM51 line. Indications of inflammatory responses were observed at 72 h, implying a delayed initiation of inflammation in the healthy line. In parallel, at 24 h, the i705-C2 line showed activation of the NFκB, PI3K, and chemokine signaling pathways, thus implying the initiation of inflammation. The expression of *NLRP3* which is known to play a role in inflammation, immune responses, and apoptosis was enhanced 1.9-fold in the i705-C2 line only [61]. The observed activation of the NFκB, MAPK, and calcium signaling pathways along with increased IL-6 and IL-8 expression in our CNS model confirms the initiation of neuro-inflammation after BF treatment in these organoids, as the observed pathways have also been reported to be associated with other neuro-inflammatory diseases [62].

On the other hand, positive regulation of CREB transcription factor activity could be a response to stress or the induction of inflammatory cascades [63]. Zhang et al. described that bilirubin can regulate the Ca^2+^ channel opening [64]. This might have a correlation with the observed activation of calcium signaling upon BF exposure in the KEGG pathway analysis. The Ca^2+^ influx into cortical neurons regulates the expression of nNOS mediated by a CREB transcription factor [64]. The positive regulation of the CREB expression was observed in the UM51 cell line. Of note, CREB is associated with inflammation and apoptotic cell death, while both could be possible effects of BF on the organoids [65]. The UM51 line showed a 1.2-fold increase in both the mRNA and protein expression of CREB and a 1.2-fold increase in phospho-CREB protein expression. Apoptotic cell death was observed in both cell lines at 24 h post-BF exposure. However, the UM51 line showed more than a 30% increase in apoptotic cell death, while the patient line showed around a 20% increase. γH2AX, which can be a marker for both DNA damage and apoptotic cell death, also underwent a 1.4-fold increase in protein expression in the UM51 line [66]. Multiple stages of necroptosis and apoptosis signaling cascades can be regulated by each other. As the necroptosis-associated genes such as *XIAP*, *TRAF2*, *TRAF5*, *STAT1*, *STAT2*, *STAT3*, *STAT4*, *JAK2*, *BIRC3*, *BIRC5*, and *BIRC6* are upregulated in the patient organoids, this might imply reduced apoptosis but increased inflammatory response due to necroptosis compared to the healthy organoids [67,68,69,70,71].

Through all the observed analyses, the UM51 line showed increased apoptotic cell death and DNA damage and repair-related gene expression at 24 h post-BF treatment and then the activation of inflammatory-related pathways at 72 h. On the other hand, the i705-C2 line did not show increased DNA damage and repair-related gene expression at 24 h (*ATM*, *ATR*, *CHEK1*, and *CHEK2*). However, the i705-C2 line leaned toward the initiation of inflammation at 24 h, which then decreased at 72 h. Overall, these observations are indicative of a switch or shift in the initiation of inflammation and cell death-related pathways in these cell lines after treating with BF. Although both cell lines showed inflammatory responses after BF treatment, they seemed to be adopting distinct pathways and time points to respond to the BF-induced stress which results in apoptotic cell death, DNA damage–repair, or inflammation.

## 5. Conclusions

In summary, this study provides valuable molecular insights into BF-induced neuro-inflammation in iPSC-derived cortical organoids. The global gene expression analyses provided an overview of the distinct pathways and genes which might be associated with the neuro-inflammatory effects induced by BF. The iPSC-derived cortical organoids employed in this study represent a Crigler–Najjar syndrome model to study defective UGT1A1 and its subsequent phenotypic manifestation and potential application for future BIND-associated toxicological studies and drug screening.

## Figures and Tables

**Figure 1 cells-12-02277-f001:**
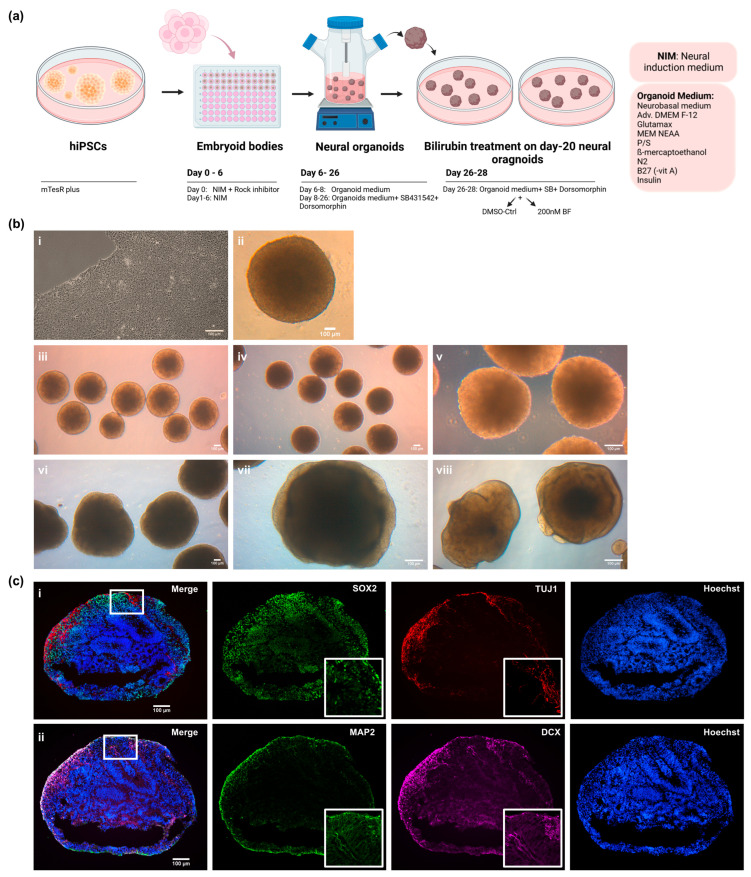
Generation of iPSC-derived cortical brain organoids and treatment with free bilirubin (BF). (**a**) Schematic depiction of the generation of cortical organoids. (Created with BioRender.com, accessed on 10 August 2023). (**b**) Bright-field images show the cortical brain organoid generation from iPSCs to organoids. Scale bars depict 100 µm. (**i**) iPSCs culture from patient-derived iPSC line (i705-C2) (**ii**). EBs were formed after 24 h of seeding in U-bottom 96-well plates. (**iii**, **iv**, **v**) Day-1 organoids (day 7 of differentiation). (**vi**, **vii**, **viii**) Day-20 organoids before BF treatment. (**c**) Neural identity of day-15 cortical organoids was confirmed by the expression of (**i**) the radial glia marker SOX2 (green) and neuronal marker TUJ1 (red), and (**ii**) MAP2 (green) and DCX (magenta), on sections and IF staining. Scale bars depict 100 μm.

**Figure 2 cells-12-02277-f002:**
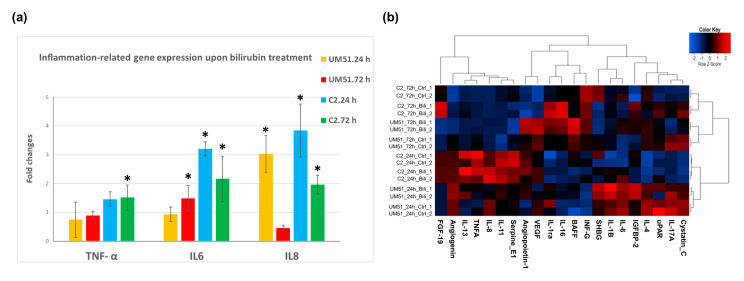
BF enhances the expression of pro-inflammatory cytokines. (**a**) qPCR analysis shows increased mRNA expression of pro-inflammatory cytokines such as *IL-6* and *IL-8* for both cell lines, while *TNF-α* was upregulated only for the patient line. Inflammatory responses were observed at 24 h in the patient line but declined gradually at 72 h, whereas the initiation of inflammatory responses shifted at 72 h for the UM51 line. Depicted values are the mean of three independent (n = 3) experiments. Error bars depict ± 95% confidence interval. Asterisk (*) depicts significance, which is determined by *p* value ≤ 0.05. Significance was calculated by using Student’s unpaired two-sample *t*-test based on a difference between each sample mean compared to the mean value of the corresponding controls. Values were normalized to *RPLP0* (housekeeping gene) and subsequently to DMSO-treated control organoids. (**b**) Cytokine array-based secretome analysis showed increased IL-6, IL-8, and TNF-α secretion along with other cytokines upon BF exposure, while the healthy control line showed lower IL-6, IL-8, and TNF-α secretion compared to the patient line.

**Figure 3 cells-12-02277-f003:**
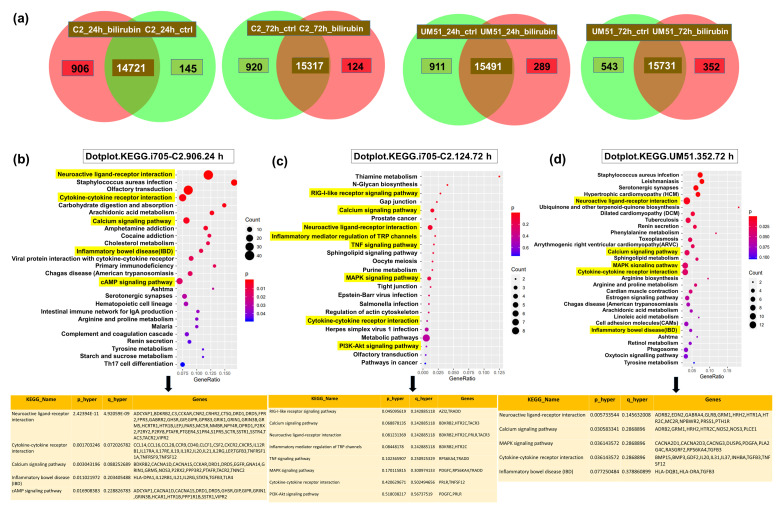
Overview of distinct activated pathways upon BF treatment in cortical brain organoids. (**a**) Venn diagrams show the uniquely and differentially expressed genes upon bilirubin treatment for UM51 and i705-C2 cell lines for both the 24 h and 72 h time points. Dot plots from KEGG-associated pathways and corresponding genes revealed cytokine–cytokine receptor activation in the patient line at 24 h post-treatment and (**b**), which goes down gradually at 72 h (**c**). This cytokine–cytokine receptor activation was not observed in the UM51 line at 24 h (Appendix A) but observed at 72 h (**d**). Dot plots show the activation of calcium signaling pathway, MAPK signaling pathway, and neuroactive ligand–receptor interaction for both cell lines at 72 h post-BF exposure.

**Figure 4 cells-12-02277-f004:**
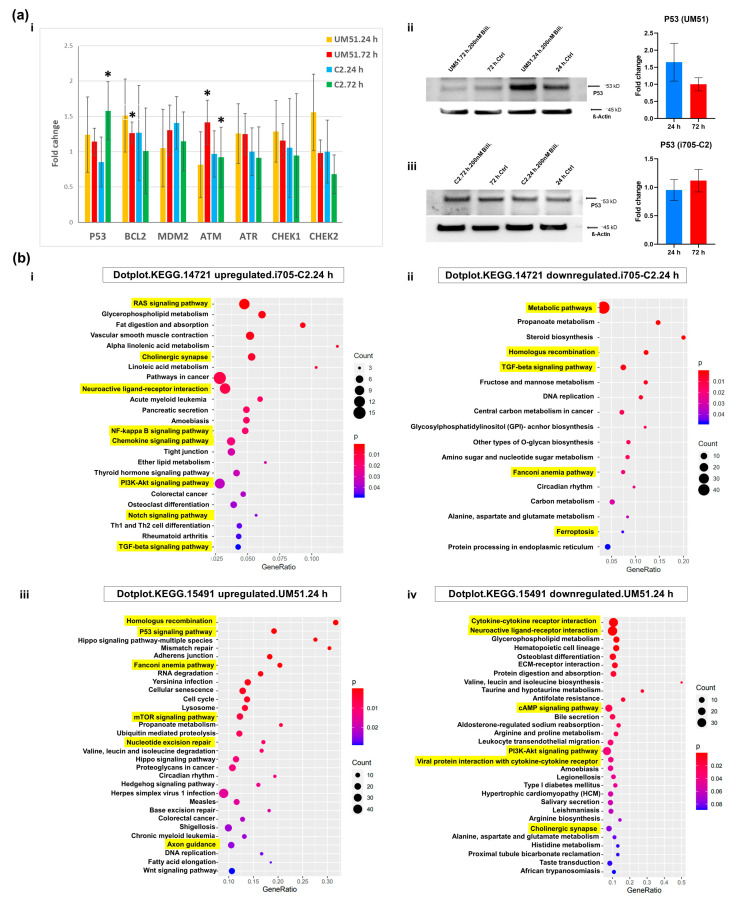
BF treatment differentially regulated DNA damage and repair-related pathways. (**a**) (**i**) qRT-PCR analysis shows DNA damage and repair-related gene (*P53*, *BCl2*, *MDM2*, *ATM*, *ATR*, *CHEK1*, *CHEK2*) expression after BF treatment. Depicted values are mean of three independent (n = 3) experiments. Error bars depict ± 95% confidence interval. Asterisk depicts significance, which is determined by *p* value ≤ 0.05. Significance was calculated by using Student’s unpaired two-sample *t*-test based on the difference between each sample mean compared to the corresponding control mean. Values were normalized to *RPLP0* (housekeeping gene) and subsequently to DMSO-treated control organoids. (**ii**, **iii**) WB analysis shows increased P53 protein expression at 24 h in the healthy line and at 72 h in the patient line, thus implying DNA damage and/or apoptotic cell death. Bar graphs show protein expression in fold change. Depicted values are mean of three independent (n = 3) experiments. Error bars depict mean ± standard deviation (SD). Values were normalized to β-Actin (housekeeping protein) and subsequently to DMSO-treated control organoids. (**b**) Dot plots from KEGG pathways revealed DNA damage and repair-related pathways such as P53 signaling pathway, homologous recombination, Fanconi Anemia pathways to be upregulated in the healthy line at 24 h (**iii**) and downregulated in the patient line (**ii**). Additionally, at 24 h, the patient line showed activation of NFκB, PI3K, and chemokine signaling pathways, thus implying the onset of inflammation (**i**). Cytokine–cytokine receptor interaction, neuroactive ligand–reactor interaction, cAMP signaling pathway were downregulated in the healthy line at 24 h (**iv**).

**Figure 5 cells-12-02277-f005:**
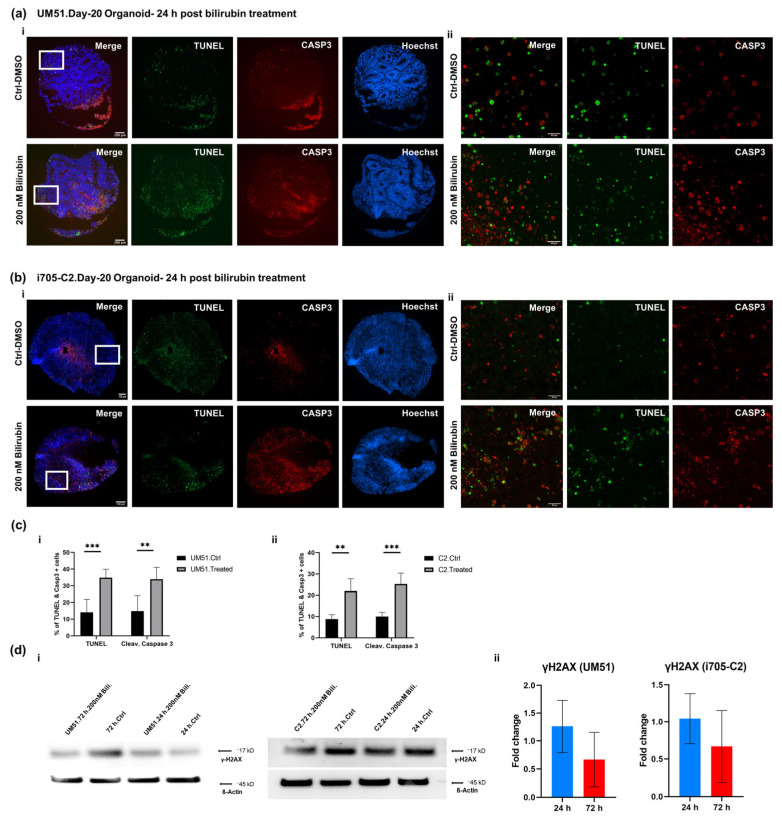
BF treatment initiates apoptotic cell death on cortical neurons. IF staining of day-20 cortical organoid sections after 24 h of BF exposure for the healthy line (**a**) and patient line (**b**). In panels (**a**) (**i**), (**b**) (**i**), (**c**) (**i**, **ii**), we see increased apoptotic cell death as confirmed by the presence of TUNEL positive (green) staining and the apoptotic cell death marker-cleaved CASP 3 (red). Scale bars depict 100 μm. (**a**) (**ii**), (**b**) (**ii**) Confocal imaging of the TUNEL (green), cleaved CASP 3 (red) positive cells shows a higher magnification. Scale bars depict 20 µm. (**c**) (**i**, **ii**) Bar graphs indicate quantification of TUNEL and cleaved CASP3 staining (n = 6). Error bars depict mean ± SD. Asterisk depicts significance, which is determined by *p* value ≤ 0.05, ** *p* < 0.01 and *** *p* < 0.001. Significance in comparison to control was calculated by using Student’s unpaired two-sample *t*-test. (**d**) (**i**, **ii**) WB analysis showed increased γH2AX expression at 24 h post-BF treatment (n = 3). Bar graphs show the protein expression in fold change. Error bars depict mean ± SD. Values were normalized to β-Actin (housekeeping protein) and subsequently to DMSO-treated control organoids.

## Data Availability

All microarray data generated and analyzed during the current study are available in the GEO repository (https://www.ncbi.nlm.nih.gov/geo/, 16 July 2023) under the accession number GSE243133.

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
