# Peer review of "Free Bilirubin Induces Neuro-Inflammation in an Induced Pluripotent Stem Cell-Derived Cortical Organoid Model of Crigler-Najjar Syndrome"

_cells, 2023, doi:10.3390/cells12182277_

Round 1

Reviewer 1 Report

The manuscript by Pranty et al. is innovative in using Pluripotent Stem Cell-derived Cortical Organoid and hepatocyte-like cells from a Crigler-Najar patient, a rare hereditary unconjugated hyperbilirubinemia disease. Though induced neurotoxicity was produced by free bilirubin addition to the culture models, the authors identified that the inflammatory signaling processes differ between the healthy individual and the patient with the Crigler Najjar syndrome. However, there are several major incorrections that should be addressed when mentioning bilirubin species and jaundice-induced neurotoxicity, as well as absence of crucial information in the Methods section. Additional data is required to better identify the neuroinflammation status induced by free bilirubin, as claimed by the authors. English editing is necessary.

Major comments

Abstract/Introduction

There are several major flaws that need to be fixed, such as:

Line 12 - Bilirubin-induced neurological damage (BIND), which is also known as Kernicterus.

The statement is incorrect. High levels of unconjugated (not albumin-bound) bilirubin known as free bilirubin potentially cause neurotoxicity and neurological damage that may progress to kernicterus, defined by a specific pattern of bilirubin deposition. Therefore, such concepts are not similar, and the Authors should rephrase the sentence. The Authors used the same wrong terminology in their previous published review in Cells 2022. Please see https://dx.doi.org/10.21037/pm-21-37.

Line 13/394 – “…resulting in unconjugated bilirubin (UCB) to cross the blood–brain barrier and accumulation.

The statement is incorrect. Only the free bilirubin crosses the blood-brain barrier, which is about 5% of the unconjugated bilirubin. Authors should rephrase the sentence.

Line 16 – Gilbert syndrome is associated with mild unconjugated hyperbilirubinemia, though it may be in the origin of neonatal jaundice, without clinical illness or kernicterus. Authors should rephrase the sentence.

Line 20 – 200 nM UCB. This reviewer thinks that the Authors mean 200 nM free bilirubin. If so, Authors should replace UCB by free bilirubin whenever referring to bilirubin in the absence of albumin.

Line 32 - Clinical jaundice – should be replaced by neonatal jaundice, once there are other forms of clinical jaundice including those caused by elevated levels of conjugated bilirubin.

Lines 35/46 - …when the unconjugated bilirubin (UCB) starts accumulating and cross the blood-brain barrier due to a defective bilirubin conjugation machinery. The same as previous. The Authors should indicate free bilirubin instead. Please see https://doi.org/10.1038/sj.jp.7211157 and https://dx.doi.org/10.21037/pm-21-37

Lines 245/394- lipophilic unconjugated bilirubin (UCB) is a disused classification. Actually, bilirubin is somehow a polar molecule that due to hydrogen bonds become insoluble in water. But also, do not solve in oil for instance. This same definition is assumed in previous publications. Papers from Donald Ostrow, Rolf Brodersen and Claudio Tiribelli have extensively addressed this issue (e.g., PMID: 7852850 and PMID: 429290, where you may find …Bilirubin is generally considered a lipophilic substance, and its neurotoxicity is ascribed to an affinity for lipids in the central nervous system. In the present paper, it is shown that the solubility of bilirubin in apolar solvents and in triglycerides is low and increases with solvent polarity. Consequently, bilirubin should not be characterized as lipophilic.).

Methods

·        Authors should indicate how the solutions of bilirubin were performed. This is very important to better understand if albumin was used, in the case UCB, or in its absence, free bilirubin. Whether NaOH or DMSO was used to solubilize bilirubin and if pH was corrected. This is also crucial to understand the use of 200 nM bilirubin. Such details are mandatory. Which commercial bilirubin was used. Was it purified? Please see PMID: 4659001.

·        Please also indicate the different cells and the treatment processes with bilirubin to make clear the study and the data achieved.

·   Please indicate the clinical data for the samples used in the study (age, sex, mutation, etc).

·        Please indicate how were the iPSCs obtained from the healthy and CNS patient. Were there obtained from fibroblasts? Also missing is the origin of the iPSCs. Characterization and the protocols used are also missing.

·        Statistical analysis assessment is missing and should be described.

Results/Discussion

·        Line 220/223 3D-cortical neuronal organoids – Are the Authors referring to 3D-cortical neural organoids? As far as I know, such spheroids may also contain oligodendrocytes and astrocytes. Please address this issue.

·        Please indicate when you are mentioning HLCs or neural cells. Line 224 if presenting UGT1A1 mutation it should be clear that you are referring HLCs and not cortical ones. Please be precise. Please also indicate in the legend the abbreviations used. Authors mention that cells retained the mutation. However, the mutation is not indicated. If the Authors did not identify the mutation should mention defective UGT1A1. However, it would be interesting if the patient has been characterized for the mutation and the clinical data presented in a Table, even if Supplementary. As shown in Fig S1 there is a reduction in the UGT1A1 gene and protein. Therefore, the sample used should be a Crigler-Najjar type II. If this is the case, please indicate as so. However, the Authors indicate a previous paper [13] where the same patient was indicated as a Crigler-Najjar I patient. Please reconcile the information and explain, once the UGT1A1 expression (Fig S1) is not absent (line 399) as claimed by the Authors, but residual as indicated for type II.

·        In the Discussion section the Authors refer to complete or partial mutation – line 393. Please explain what you mean by such classification. These aspects should be clarified by the Authors.

·        Line 235 (a) Schematic depiction of the generation of cortical organoids. Why is bilirubin treatment indicated in Fig. 1? Why is it included such an addition? And why only here we see that DMSO may have been the bilirubin solvent. If so, it seems that the experiments were performed with free bilirubin and not UCB that usually is applied for bilirubin binding to albumin. In which percentage was added to cells? This is important once the percentage of DMS should be below 5%. Such concepts are very important, require correction and added information. The scheme in (a) should be indicated separately from the other images in Fig 1.

·        Why the Authors indicate magnifications in Fig. 1 and include them in the images. Either one or another way should be used. If using those of Figures, please increase font size in (d).

·        Data from section 3.2 indicate that some cell death may have occurred in incubations for 72 h, what then will justify the decrease observed in IL8 and even IL6 in the patient. This reviewer recommends such evaluation at both 24 and 72 h. The Authors only present apoptotic data at 24 h and should include results also at 24 h. However unconjugated and not albumin-bound bilirubin is known to also trigger the necrosis of cells, and this determination is missing (e.g., Fluoro-jade stain). Did the Authors check the expression of IL8 receptors?

·        Font size should be increased in Fig. 2b for better reading, as well as in many other Figures.

·        IL1-ra is a receptor and a natural inhibitor of the pro-inflammatory effect of IL1β. Please comment on this.

·        Please correct and indicate the meaning of the abbreviations in all the text.

·        Please also do not mention UCB if you are using a solution of bilirubin in DMSO. Whenever UCB is used it is assumed that albumin is present and mimics the circulation in serum/plasma. Please, correct UCB in all the MS as free bilirubin, usually designated as Bf, if this reviewer has well interpreted the treatment.

·        The Authors claim the presence of neuroinflammation by the treatment of cortical organoids with bilirubin. It will be important to identify and characterize the presence of glial cells, such as astrocytes, considering their relevance in the production of cytokines. It would be important to also stain the neuronal cultures with non-neuronal markers, e.g., GFAP, to demonstrate the purity of the neuronal population. Please add such data.

·        Line 322 - The Authors claim that “The mRNA expression of NLRP3 was enhanced only for the i705-C2 line- seems to be enhanced would be more accurate.

·         Line 417 – the Authors suggest that “This observation might imply that inflammatory responses were initiated earlier in the i705-C2 line and gradually reverted to normal levels with time as a consequence of cellular defense mechanisms, establishing homeostasis”. To reinforce the statement, it would be interesting to show IL-4 and IL-10 (indicated as anti-inflammatory cytokines) at both 24 and 48 h incubation.

·        Line 532 – please comment on the sentence However, the UM51 line showed more than 30% increase in apoptotic cell death, while the patient line showed around 20% increase. Could that be due to necrosis in the patient cell line? As mentioned, it would be interesting to add data on cell death by necrosis at both 24 and 72 h of bilirubin treatment in the health and patient cell lines.

 English editing is required, and several typos along the text need to be corrected.

 Minor comments

·        embroid bodies – should be embryoid bodies – line 92;

·        EBs abbreviation in line 93 should be introduced above;

·        hepatocyte-like cells (HLCs) is only indicated in line 400. Please refer to when indicated by the first time – line 106;

·        immunofluorescnece – line 118 should be immunofluorescence;

·        TritonX – line 120 should be Triton X-100;

·        XLCytokine Array – line 164 should be XL Cytokine Array;

·        S1 legend faded and not fade.

·        Supplementary Fig. 1 Legend - The statement “The UGT1A1 was only expressed on the healthy line” does not reflect the data. Please correct.

·        Supplementary Fig 5 – The Figure shows both yH2AX and Ki67, but the Authors refer only to the first. Please correct. Confocal images quantification would add significance to the statement.

 English editing is required, and several typos along the text need to be corrected.

Author Response

We would like to convey our gratitude towards the reviewer for his/her valuable comments, which have improved the quality of our article. We have included and answered all the comments addressed by the reviewer. To be noted, the reviewer had suggested an experiment which we could not perform due to delivery issues and delayed necessary reagents. However, we have addressed the question using our transcriptome data to analyse regulated expression of the genes associated with necroptosis as illustrated in the KEGG pathway database. 

Please see the attached pdf file for the addressed responses.

Please find the attached powerpoint file, and raw file, that we have additionally attached to answer some of the points.

Reviewer 2 Report

Authors in the present study used hiPSCs-derived organoids to model Bilirubin-induced neurological damage in vitro and studied the molecular basis of the effects of unconjugated bilirubin in developing human brain. The article was well written, designed, presented, and the conclusion is evidence based. I recommend acceptance in the present form.

Author Response

We  would like to convey our gratitude towards the reviewer for reviewing our manuscript.

Reviewer 3 Report

This manuscript addresses an important topic by employing human stem cell-derived organoids. The neurotoxicity of unconjugated bilirubin can occur in neonates under certain conditions, one of them is caused by mutation in UGT1A1. The approach is straightforward, but not in all instances explicitly described. The findings examplify the high potential of iPSC-derived disease modeling, however, to improve its impact, i have a few suggestions:

-Neurotoxicity in general involves neuroinflammation and apoptosis.  It is not evident why the pathways identified are specific for kernicterus. Hence, I suggest to compare the CNS organoids not only with wildtype organoids, but also with other neurotoxic or well-chosen neurodegenerative diseases. I hold this point to be highly important to increase the meaningfulness and underline the importance of the study. NFkb, TGFbeta etc. could be markers of many scenarios involving cellular damage.

- The discussion should be shortened and focussed on the findings rather than repeating aspects of the introduction.

-Hoechst staining is one of the standard stinings in stem cell research. Please correct the term in figure 5 and elsewhere.

- Please avoid terms with emotional connotations in scientific manuscripts (e.g. "meagre" line 72.)

- Please explain all abbreviations upon first mentioning.

The English could benefit from editing by a native speaker.

Author Response

We would like to convey our gratitude towards the reviewer for his/her valuable comments, which have improved the quality of our article. We have included and answered all the comments addressed by the reviewer. 

Reviewer 4 Report

Authors describe the use of hiPSCs organoids to investigate the effects of unconjugated bilirubin on the developing human brain, with a focus on Bilirubin-induced neurological damage. The study highlights the potential of hiPSC derived 3D-brain organoids as a platform for studying the etiology of BIND. The study also discusses methods such as cytokine assays and microarray analysis to investigate changes in gene expression and pro-inflammatory cytokine expression in response to bilirubin treatment.

The study involved global gene expression analysis and provided insight into distinct pathways and genes associated with neuro-inflammatory effects caused by bilirubin. The study also included a Crigler-Najjar syndrome model to study UGT1A1 mutation and its phenotypic manifestation. The results showed distinct pathways and time points for responding to bilirubin -induced stress resulting in apoptotic cell death, DNA damage repair, or inflammation in both the healthy and patient lines.

Major comments:

Q1. I have found only software environment titles, but no description of what statistical methods were used to determine whether changes in expression are statistically significant. Was it a t-test or a significance analysis of microarrays ?

Q2. Every time authors say that some mRNA expression or apoptotic cell death increases or decreases, there is no information on the statistical significance of the observations. Without this information, it is impossible to say if the changes were significant.

Q3. What cell types in hiPSC derived organoids are responsible for IL-6 and IL-8 cytokines secretion?

Minor comments:

Q1. Line 166. Please clarify what equipment was used to scan cytokine assay.

Q2. Author mention that organoids were studied in triplicates (lines 144, 224, 247). Does this mean, the number of measurements n=3 ?

Author Response

(The authors gave the same response as above.)

Round 2

Reviewer 1 Report

The authors have satisfactorily addressed most of my questions and concerns, thus improving the quality of the manuscript. I consider the manuscript acceptable for publication.